# Incomplete human reference genomes can drive false sex biases and expose patient-identifying information in metagenomic data

Caitlin Guccione[1,2,3,23], Lucas Patel[2,3,4,23], Yoshihiko Tomofuji[5,6,7], Daniel McDonald[3], Antonio Gonzalez[3], Gregory D. Sepich-Poore [8,9], Kyuto Sonehara [5,6,7], Mohsen Zakeri[10], Yang Chen[3,11,15], Amanda Hazel Dilmore[3,11], Neil Damle[12,13], Sergio E. Baranzini [14], George Hightower[15,16], Teruaki Nakatsuji [15], Richard L. Gallo [15,17], Ben Langmead [10], Yukinori Okada [5,6,7,18,19], Kit Curtius [1,20,21,24] ✉ & Rob Knight[3,8,12,17,22,24] ✉

As next-generation sequencing technologies produce deeper genome coverages at lower costs, there is a critical need for reliable computational host DNA removal in metagenomic data. We find that insufficient host filtration using prior human genome references can introduce false sex biases and inadvertently permit flow-through of host-specific DNA during bioinformatic analyses, which could be exploited for individual identification. To address these issues, we introduce and benchmark three host filtration methods of varying throughput, with concomitant applications across low biomass samples such as skin and high microbial biomass datasets including fecal samples. We find that these methods are important for obtaining accurate results in low biomass samples (e.g., tissue, skin). Overall, we demonstrate that rigorous host filtration is a key component of privacy-minded analyses of patient microbiomes and provide computationally efficient pipelines for accomplishing this task on large-scale datasets.

Metagenomic next-generation sequencing (mNGS) encompasses various high-throughput DNA profiling techniques that enable environment-agnostic taxonomic profiling of microorganisms, including bacteria, archaea, fungi, and viruses[1]. mNGS has shown increasing adoption in clinical contexts for diagnosing infectious diseases, surveilling microbial pathogens, and predicting antibiotic efficacy[2,3] across fecal, skin, and tissue samples. Utilizing mNGS in these settings is attractive due to its untargeted and high-throughput characteristics; however, its untargeted nature can result in substantial and confounding amounts of non-microbial DNA (e.g., human DNA) when processing low-microbial biomass samples, especially at higher sequencing depths.

To mitigate the influence of non-microbial DNA on metagenomic studies, diverse host depletion techniques have been developed, ranging from experimental modification of DNA extraction steps (e.g., differential lysis)[4] to real-time sorting of reads during sequencing[5]. Computational host filtration, or simply host filtration, refers to computational approaches for removing host DNA from sequenced samples, regardless of whether prior host depletion steps were performed. Separating host genetic information from microbial counterparts is a crucial step in mNGS workflows, especially in the analysis of low microbial biomass samples, such as those derived from skin, saliva, or tumors[6,7]. Additionally, these methodologies are imperative to

increase the rigor with which debated microbial communities, like putative blood-borne microbiota, can be assayed.

When human DNA reads from mNGS are not correctly identified during the host filtration step, they may be incorrectly classified as microbial reads, creating potentially misidentified taxonomic classifications and biased effect sizes (inflation or deflation of mismapped taxa)[7]. These misclassifications can manifest as false positive taxonomic classifications, and in this work we further demonstrate these false positives can alter biological conclusions. Further, failure to remove human DNA from metagenomic sequencing samples can leak private genetic information about the host into putatively microbial data, enabling re-identification of study participants[8].

Although host filtration is generally a common preprocessing step[9], the algorithmic choice for host filtration and employed human reference database(s) can result in substantially different biological results[7]. Tools differ between pipelines, but most host filtration approaches map reads to a host reference genome followed by sequence-based computational subtraction of host reads to obtain human-filtered data[10].

Most host filtration tools[10] and recommended host filtration workflows[11,12] exclusively use a single human reference, which fails to capture the diversity of human genomes and cannot remove population-specific variation. Portions of the human genome that are incomplete in these references, such as the Y chromosome in GRCh38 or earlier versions of T2T-CHM13 (v1.0), can permit flow-through of human reads from those regions to microbial mapping steps, leading to the mismapping of taxa during classification and artifactual data distributions (e.g., false sex differences in the low biomass microbial profiles). Moreover, regions of population-specific genome variation or haplotypes not well covered in singular reference genomes can allow leakage of patient-identifying information in microbial reads[8]. To date, previous work[13] has either failed to incorporate pangenome references[14] or to provide computationally efficient methods capable of host filtering across dozens of human genomes[7], and both are needed to protect patient privacy and improve output quality. Therefore, we were motivated to explore more efficient methods for host filtration using the most comprehensive human references available to protect the privacy of disseminated metagenomics datasets and mitigate artifactual biases associated with missing genome regions.

In this work, we identify and resolve an artifactual technical effect caused by insufficient host filtration in quantifying the microbial profiles associated with tumor tissue from mNGS. We then implement and benchmark three improved host filtration approaches that leverage two complementary algorithmic approaches and a wide variety of human reference genomes to maximize host read removal. We apply these novel methodologies towards multiple sample types in both low and high biomass conditions. Finally, we show that improved host filtration prevents host re-identification from mNGS datasets with matched genotyping information. These efforts support utilization of comprehensive host filtration preprocessing for current and future mNGS studies to increase data robustness and protect patient privacy.

## Results

### Artifactual sex splitting across metastatic cancers

We initially and incidentally discovered detrimental effects from improper host filtration when exploring sequencing data from a cohort of metastatic human tumor tissues (Hartwig Medical Foundation[15] hereafter "HMF"). These data were originally processed and published before the release of T2T-CHM13v2.0[16], which added a complete human Y chromosome, and have since been independently analyzed for microbiomes[17]. After isolating non-human reads from deep, whole-genome sequenced samples of 4902 metastatic tumors (Supplementary Data 1), we applied quality and length filtering, followed by re-alignment against GRCh38.p7. Surprisingly, our initial

analysis of putative metastatic tumor low biomass microbial profiles revealed significant differences between male and female-labeled samples (Fig. 1a; $p = 0.00025$, RPCA[18]-PERMANOVA). Subsequent re-analysis of the same data using T2T-CHM13v2.0[16], which included the first complete Y chromosome, abolished the male-female sex separation in our data (Fig. 1b; $p = 0.29$, RPCA[18]-PERMANOVA). These results suggested that missing regions of human reference chromosomes can directly cause related artifactual biases in downstream microbiome data.

To validate that this result was not unique to RPCA, we re-calculated results with Weighted and Unweighted UniFrac[19], Bray-Curtis[20] dissimilarity, and Jaccard[21] similarity index. We also tested another microbial database, Web of Life release 2[22], all with and without rarefaction. Notably, neither the choice of rarefaction level nor microbial database affected the identification of artifactual sex differences. The results were less affected by filtration against T2T-CHM13v2.0 when *qualitative* distance metrics were used (i.e., Unweighted UniFrac and Jaccard similarity index). However, quantitative metrics (i.e., Weighted UniFrac and Bray-Curtis) reproduced our original findings that additional filtration against T2T-CHM13v2.0 abolished sex differences (Supplementary Data 2).

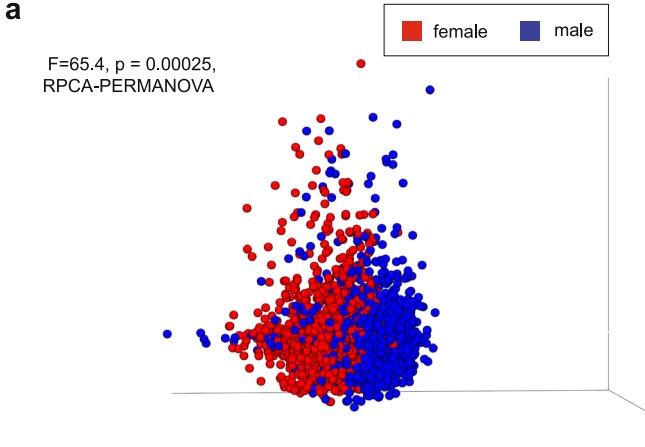

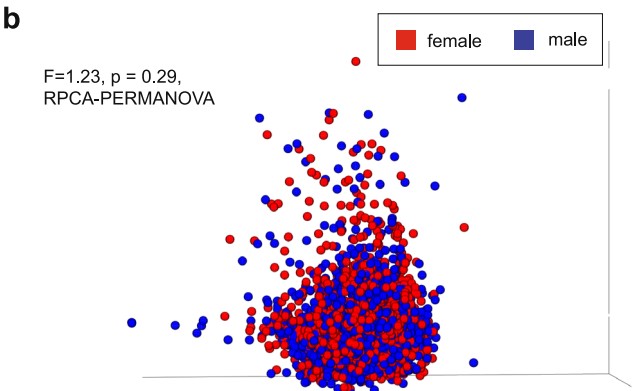

**Fig. 1 | Sex biases identified in inadequately host-filtered human tumor tissue data. a** RPCA of microbial relative abundance quantification from tumor samples in the Hartwig Medical Foundation Database, which was originally subject to GRCh38.p7 filtration exclusively. Statistically significant differences were found between male and female groups (PERMANOVA; pseudo-F = 65.4, $p = 0.00025$). **b** Identical dataset and pre-processing steps done in **a** but with the addition of the T2T-CHM13v2.0 reference genome in host filtration. Differences were not statistically significant between male and female groups (PERMANOVA; pseudo-F = 1.23, $p = 0.29$).

To investigate why quantitative metrics were affected but qualitative metrics were not, we examined a subset of 100 samples from HMF. We randomly selected metastatic tissue samples across various body sites, all of which had metastasized from a primary colorectal tumor. We isolated reads from these samples that were removed when filtered with T2T-CHM13v2.0 but retained when filtered with GRCh38.p7. We refer to these reads as "T2T-filtered". Using Woltka and RefSeq release 200, we mapped the T2T-filtered reads to their corresponding operational genomic units (OGUs)[23] (Supplementary Data 3). Notably, 99.895% of the reads (5,590,189/5,596,038) matched only four taxa: *Toxoplasma gondii* (G000006565), *Bifidobacterium tibiigranuli* (G009371885), *Alcanivorax hongdengensis* (G000300995)*, and Tetrasphaera japonica* (G001046855). When reads we observed as being mistakenly mapped to microbial taxa are removed with T2T-CHM13v2.0 filtering, the quantitative diversity metrics more accurately reflect the per-sample microbial diversity (Fig. 1b). However, qualitative metrics relying on presence-absence (and not abundance) are less affected, and thereby more robust to, falsely inflated abundances of *T.gondii*, *B. tibiigranuli*, *A. hongdengensis*, and *T. japonica*. These results demonstrate the importance of validating conclusions with *qualitative* metagenomic methods when using older or incomplete genome references—although the human references are now essentially complete, the same is not true for other species, so this category of validation will continue to be important into the future.

To test whether these four taxa shared regions of sequence similarity with the human genome, we took the same T2T-filtered reads from the subset of 100 HMF samples described above, and mapped those sequences against RS210-clean, a version of RefSeq release 210 (2022-01-01) in which regions of microbial genomes shared with human genomes based on Exhaustive[1] and Conterminator[24] were masked. We found that using a human-scrubbed microbial database eliminated some of the T2T-filtered reads from mapping to microbes (Supplementary Data 4). 5,596,038 T2T-filtered reads mapped to microbes using RefSeq release 200, but only 53 mapped to microbes using RS210-clean. This result suggested that a cleaned microbial database may alone abolish the false male-female sex difference. However, a strong human-filtration pipeline that uses T2T-CHM13v2.0 would have removed all these T2T-filtered reads, so that none of them would map to microbes regardless of which database was used.

To confirm that a cleaned microbial database would abolish the sex difference without any additional filtration, we then applied RS210-clean on a larger subset of 477 metastatic tissue samples across various body sites, all of which had metastasized from the colon using GRCh38.p7 filtration alone. Importantly, we found that the sex differences were eliminated ($p = 0.14$, RPCA[18]-PERMANOVA). These results indeed demonstrate that host filtration of either the reads (prior to mapping) or the microbial database is sufficient to prevent sex biases.

To verify that the T2T-filtered reads from the subset of 100 HMF samples described above was in fact derived from the human Y chromosome, we aligned the T2T-filtered reads using minimap2[25] (v. 2.26) and an index based only on the Y chromosome portion of T2T-CHM13v2.0, and observed an overall alignment of 88.99% to the Y chromosome. We speculate that the remaining 11.01% are likely due to additions of regions in other chromosomes with the T2T-CHM13v2.0 release. We spot-checked alignments of a subset of Y chromosome mapped reads using the BLAST[26] web portal. For example, a read that aligns with 100% identity to the Y chromosome is also identified as *T. gondii* at 98.67% identity when using Nucleotide BLAST[26]. To confirm the T2T-filtered reads were likely mismapped to microbial genomes, we calculated the depth and breadth for the top ten multi-mapped organisms (Supplementary Fig. 1a). We observe a large coverage peak within each genome with low mean coverage depth, suggesting an artifactual signal. We extracted the genomic regions corresponding to the coverage peak for each organism and confirmed they correspond to low complexity regions of each respective microbial reference

genome (Supplementary Data 5). Finally we include coverage depth and breadth assessments for these same reads against the Y chromosome from the T2T-CHM13v2.0 and note a more uniform distribution, suggesting the true origin of the reads corresponds to the more complete Y chromosome in T2T-CHM13v2.0 rather than any microbial genome (Supplementary Fig. 1b). Overall, these data suggest that the sex differences identified in the HMF dataset are attributable to human Y chromosome sequences leaking through the GRCh38.p7 filter, which were subsequently mapped to microbial taxa containing genomic regions common to the human genome.

Inspired by this resolution to the problem of artifactual sex-specific differences, we sought to create and evaluate pipelines for thorough host filtration in a computationally efficient manner (described below). These pipelines can be conservatively combined with microbial database cleaning/masking approaches, as we and others have described elsewhere[7,24]. However, we caution that microbial database masking alone may not adequately address patient re-identification concerns, because human reads remain mixed with microbial reads, as addressed later in this work.

## Improved host filtration approach and validation

We thus proposed and benchmarked three methods for improved host filtration that utilizes traditional sequence alignment[25] and a novel indexing-based approach called Movi[27]. We evaluate multiple human references, including the most updated versions of GRCh38.p14, T2T-CHM13v2.0[16], and HPRC-2023 release[14], to maximize captured human genomic diversity. Our methods are as follows: 1) Alignment with minimap2 to GRCh38.p14 and T2T-CHM13v2.0, and indexing with Movi to GRCh38.p14, T2T-CHM13v2.0, and HPRC, 2) Alignment with minimap2 to GRCh38.p14, T2T-CHM13v2.0, and HPRC, and indexing with Movi to GRCh38.p14, T2T-CHM13v2.0, and HPRC, 3) Indexing with Movi to GRCh38.p14, T2T-CHM13v2.0, and HPRC (Fig. 2a; see Methods for details). Additionally, we compared our methods to the only other publication using HPRC for host filtration, which used all three human genome reference sets with minimap2[25] in both paired-end and single-end mode. We also benchmarked our methods against the strict host filtration method described by Sepich-Poore et al.[7] (Supplementary Fig. 2a).

To compare the run time of host filtration methods, we simulated data of 50% human and 50% microbial reads using ten sampled genomes from HPRC and over 800 complete bacterial assemblies from the FDA-ARGOS database[28]. Using these ten simulated datasets, we subsampled them at ten thousand, 1 million, and 10 million reads, followed by applying all filtration methods on each to assess their scalability (Fig. 2b). Methods 1 and 3 had comparable runtimes: 11.13 min and 11.15 min at 1 million reads, respectively. In comparison, Method 2's use of HPRC alignment with minimap2 created exponentially increasing run times (46 min, 55 min, and 2.5 h at ten thousand, 1 million, and 10 million reads, respectively) as the dataset size increased. The strict host filtering method described by Sepich-Poore et al.[7] also took the longest to complete, or 1.59 h for 1 million reads (Supplementary Fig. 2b).

We next applied the three host filtration methods to assess sensitivity on the aforementioned ten samples of 1 million reads each, excluding the ten pangenomes we used to simulate the human data during filtration. An ideal host filtration method would result in zero remaining human reads (Fig. 3a, Supplementary Data 6) and a minimal number of lost microbial reads (Fig. 3b, Supplementary Data 6). For the remaining human reads (Fig. 3a), we found significant differences between Method 1 and Method 3, as well as Method 2 and Method 3 (Wilcoxon signed-rank test, $p = 0.0020$), indicating that the combination of alignment and indexing-based approaches for host filtration outperforms indexing based approaches alone. For microbial reads lost (Fig. 3b), we found significant differences across all three methods (Wilcoxon signed-rank test, $p = 0.0020$ for all comparisons). We find that the indexing-based host filtration approach alone (Method 3)

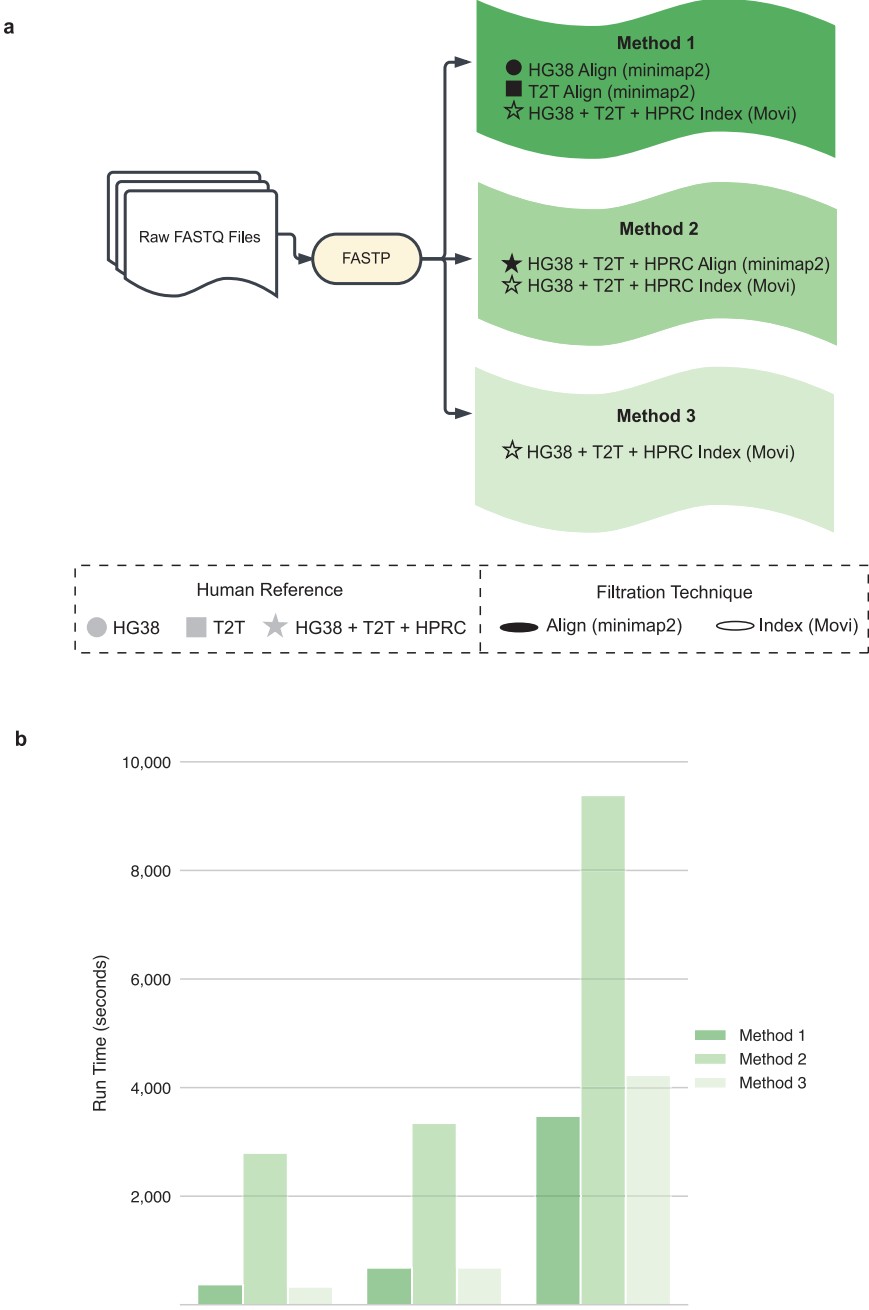

**Fig. 2 | Host filtration pipeline and runtime evaluation. a** Pipeline of host filtration methods. **b** Using simulated data with a 50/50 mix of human data from HPRC and microbial data from FDA-ARGOS, we applied the 3 host filtration methods with 3 different sample sizes. Runtimes were averaged across 10 runs per sample size. HG38: GRCH38.p14, T2T: T2T-CHM13v2.0, HPRC: Human Pangenome Reference Consortium 2024 release.

retains the greatest number of microbial reads, while alignment-based steps, as in the initial steps of Method 1 and Method 2, inadvertently discard an increasing number of microbial reads proportional to the number of human references used for alignment. Although Method 2 was most effective at removing human reads, it also removed 242.5 and 288.5 more microbial reads on average compared with Method 1 and 3, respectively. In contrast, Method 3 maximized the number of microbial reads kept, losing only 43.5 microbial reads on average, but also allowed an average of 4.5 human reads through. Method 1 struck a balance, losing 89.5 microbial reads on average and eliminating all the human reads in 8 out of the 10 cases. We found that the prior Sepich-Poore et al. [7] method performed identically to Method 2 regarding the number of human reads removed (Supplementary Fig. 3a) and unnecessarily removed an additional ten microbial reads (Supplementary Fig. 3b). Because host filtration is used in a wide range of applications, it is crucial to allow users to choose between methods and determine if, for a given application and regulatory environment, it is acceptable to lose more microbial reads while ensuring maximum human read removal; conversely, one may want to maximize the number of microbial reads retained while still removing the majority of host reads. We note that microbial reads may be lost inadvertently due to sequence similarity between microbial input reads and human reference databases when using both alignment and indexing-based approaches (see Fig. 3 and Supplementary Fig. 4).

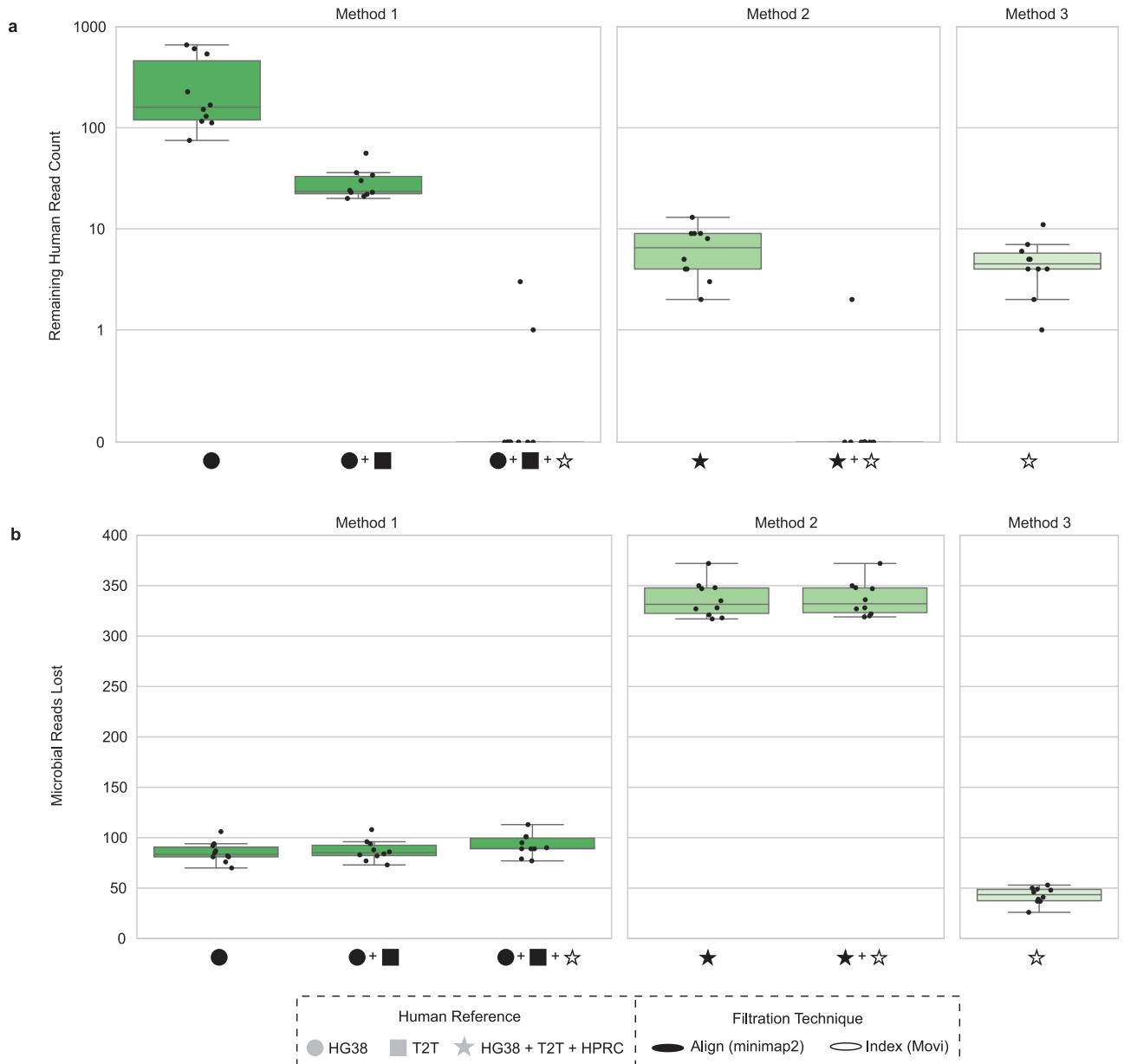

**Fig. 3 | Host filtration pipeline simulated data validation.** Using the 10 simulated datasets of 1 million reads as described in Fig. 2b, we **a** calculated the number of human reads remaining, and **b** number of microbial reads remaining, for host filtration Methods 1–3 (HPRC host filtration performed excluding the 10 genomes used for data simulation). HG38: GRCH38.p14, T2T: T2T-CHM13v2.0, HPRC: Human Pangenome Reference Consortium 2024 release. Box plots show the median (center line), interquartile range (IQR; Q1–Q3; box), whiskers extending to Q1 − 1.5 × IQR and Q3 + 1.5 × IQR, minimum and maximum values at whisker ends, and points representing individual observations both within and beyond the whisker range.

## Application of host filtration methods to low and high microbial biomass samples

To determine the robustness of these three methods across a range of microbial biomasses, we evaluated each method on human exome data as well as tissue, skin, and fecal metagenomic samples. First, we obtained 30 International Genome Sample Resource (IGSR) phase 3 human exome sequencing samples[29], which are putatively human. After sampling 1 million reads each, we examined the number of human reads remaining, with an ideal host filtering method having zero reads left. We found Method 2 left the smallest amount of human exome reads followed by Method 1, then Method 3 (average reads remaining; Method 1: 32.66, Method 2: 24, Method 3: 351.53). There were significant differences between Method 1 and Method 2 (Wilcoxon signed-rank test, $p = 2.6e−05$), between Method 2 and Method 3 (Wilcoxon signed-rank

test, $p = 8.2e−06$), and between Method 1 and Method 3 (Wilcoxon signed-rank test, $p = 3.8e−05$) (Fig. 4a, Supplementary Data 6). Mirroring the distributions seen in human simulated data benchmarks (Fig. 3a), Method 2 removed the largest number of human sequences, followed by Method 1, then Method 3. Interestingly, we found nearly ten times as many human exome reads remained compared to the simulated human data (Fig. 3a). However, without access to the samples, it is not possible to determine whether the increased number of reads in the human exome data compared to the simulated human data is due to real microbial presence (contamination or biological) in the exome sample, imperfect amplification or selection chemistry, and/or reduced performance of the host filtration procedure.

Using these three host filtration methods, we re-analyzed the aforementioned 100 colorectal tissue tumor samples from HMF, finding

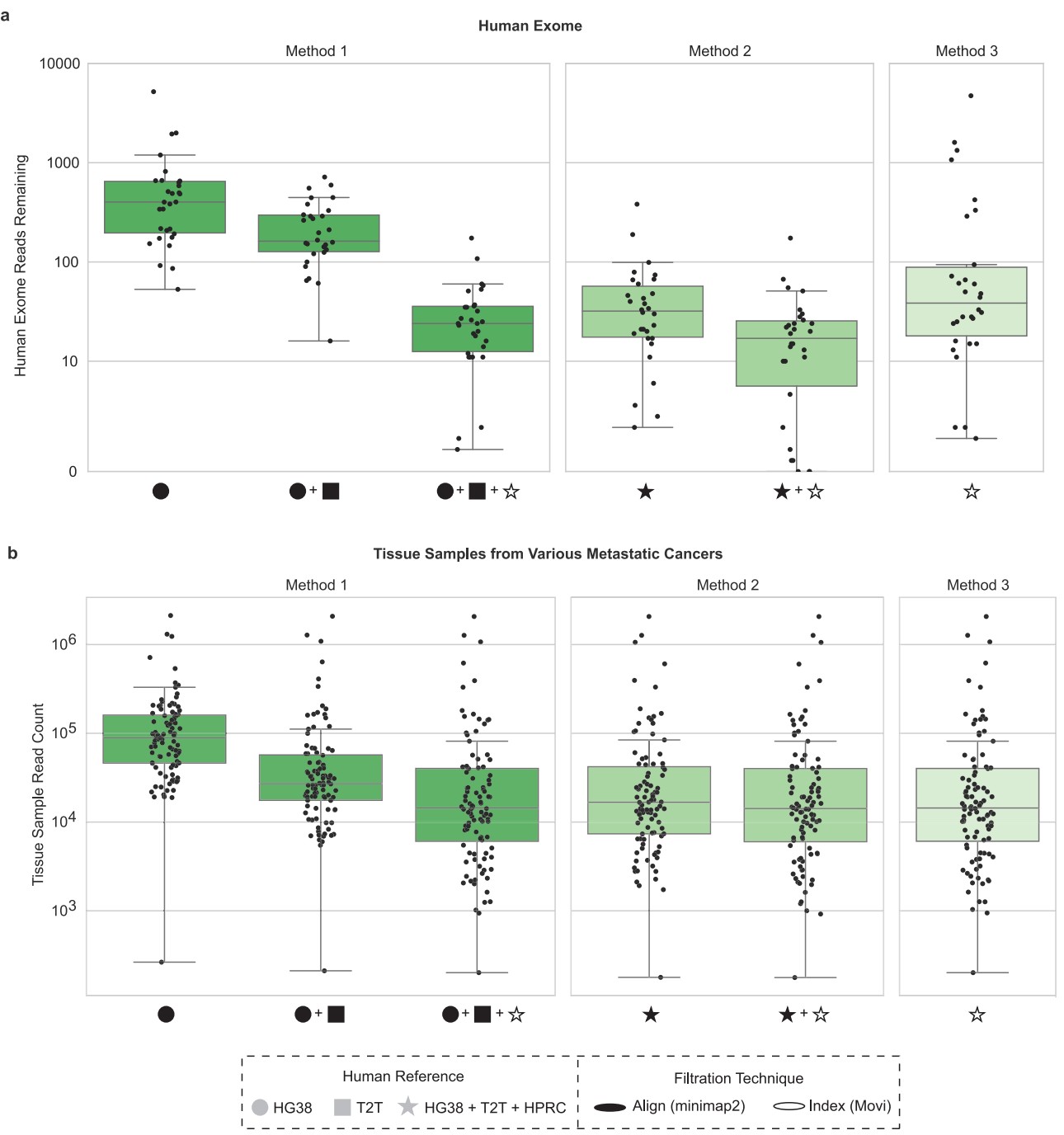

**Fig. 4 | Comparing human exome and tumor tissue samples across host filtration methods. a** The number of reads remaining after host-filtering 30 human exomes subset to 1 million reads across methods. **b** 100 metastatic colorectal cancer tissue samples were selected from HMF and read counts were calculated following application of improved host filtration methods. HG38 GRCH38.p14, T2T T2T-CHM13v2.0, HPRC Human Pangenome Reference Consortium 2024 release. Box plots show the median (center line), interquartile range (IQR; Q1–Q3; box), whiskers extending to Q1 − 1.5 × IQR and Q3 + 1.5 × IQR, minimum and maximum values at whisker ends, and points representing individual observations both within and beyond the whisker range.

additional human reads removed compared to T2T-CHM13v2.0 alone (Fig. 4b, Supplementary Data 6). For HMF total read count following host filtration, we found significant differences between Method 1 and Method 2 (Wilcoxon signed-rank test, $p = 3.9e{-}18$), between Method 2 and Method 3 (Wilcoxon signed-rank test, $p = 1.2e{-}17$), and between Method 1 and Method 3 (Wilcoxon signed-rank test, $p = 3.9e{-}18$). Again, Method 2 has the least reads followed by Method 1 and then Method 3 (average reads remaining; Method 1: 84,663.12, Method 2: 84,009.03, Method 3: 84,692.71). Although we cannot verify if the remaining reads are all microbial, we can conclude, based on the simulations, that

Method 2 likely has lower read counts due to removal of true microbial reads.

Next, we applied our host filtration methods to mNGS data from skin samples, where microbial and human DNA would be expected in varying proportions. Specifically, we analyzed 77 skin swab samples from pediatric healthy controls and subjects with atopic dermatitis (Fig. 5a, Supplementary Fig. 5, Supplementary Data 6). The percentage of non-human reads remaining across skin samples varied, consistent with distinct levels of host background within each sample, with Method 2 providing the lowest percentage of reads remaining,

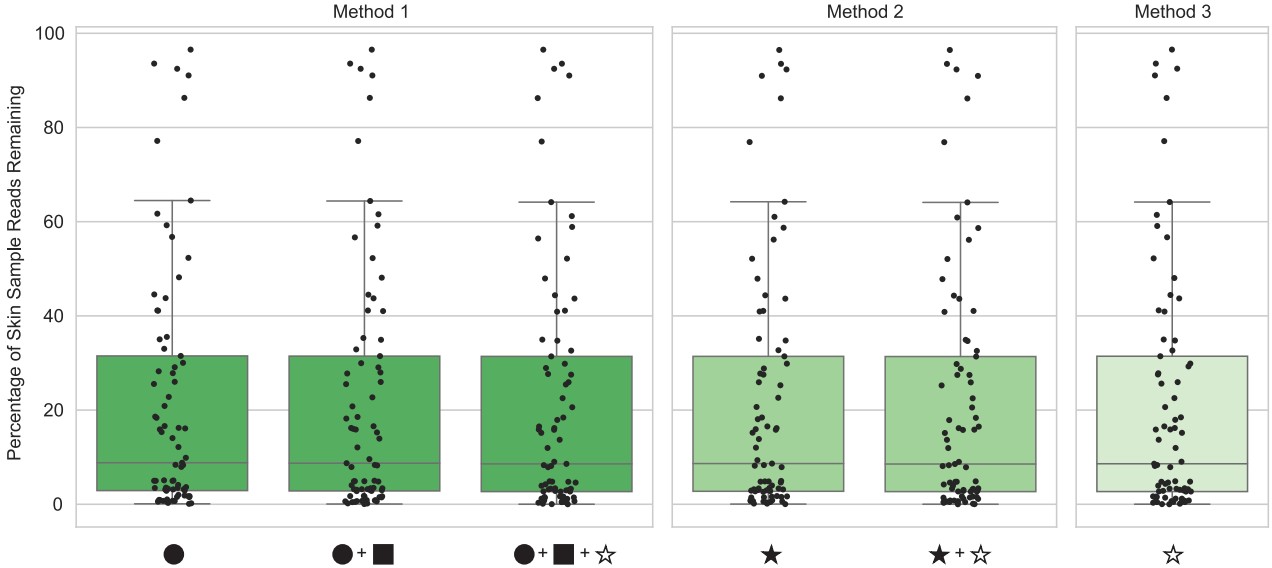

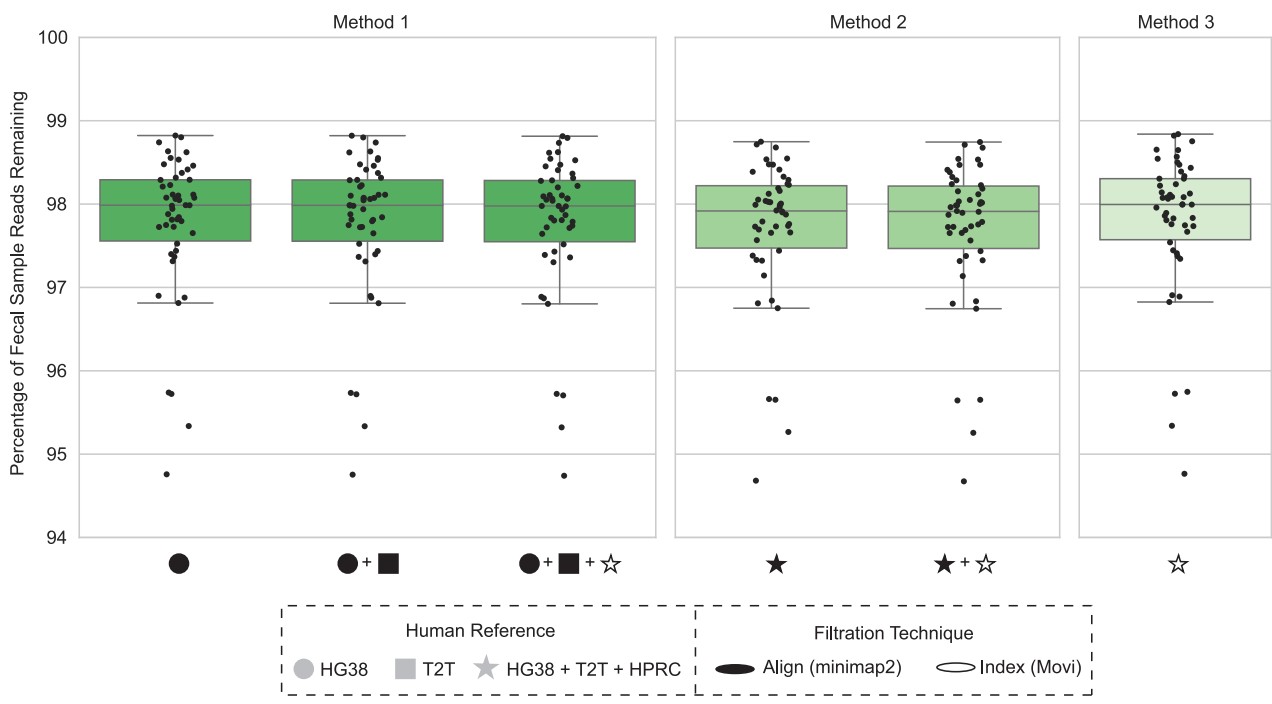

**Fig. 5 | Comparing human skin and fecal samples across host filtration methods. a** 87 human skin samples were host-filtered with the improved methods, we then calculated the percentage of reads remaining. **b** We calculated the percentage of reads remaining on a per-sample basis for each of the 50 human fecal samples examined. HG38: GRCH38.p14, T2T: T2T-CHM13v2.0, HPRC: Human Pangenome Reference Consortium 2024 release. Box plots show the median (center line),

interquartile range (IQR; Q1–Q3; box), and whiskers extending to Q1 − 1.5 × IQR and Q3 + 1.5 × IQR. Box plots show the median (center line), interquartile range (IQR; Q1–Q3; box), whiskers extending to Q1 − 1.5 × IQR and Q3 + 1.5 × IQR, minimum and maximum values at whisker ends, and points representing individual observations both within and beyond the whisker range.

followed by Method 1, then Method 3. For the total percentage of reads remaining of these skin samples following host filtration, we found significant differences across all three methods (Wilcoxon signed-rank $p = 2.5e{-}14$ for all comparisons).

Lastly, we evaluated a high microbial biomass dataset of 50 fecal samples from older adults consisting of healthy controls and subjects

with Alzheimer's disease[30] (Fig. 5b, Supplementary Data 6). As expected, we observed nominal reductions in the percentage of total reads, although still greater than 1% of reads. For the total percentage of reads remaining in these fecal samples following host filtration, we found significant differences between all three methods (Wilcoxon signed-rank test, $p = 7.6e{-}10$ for all comparisons), and the same pattern of

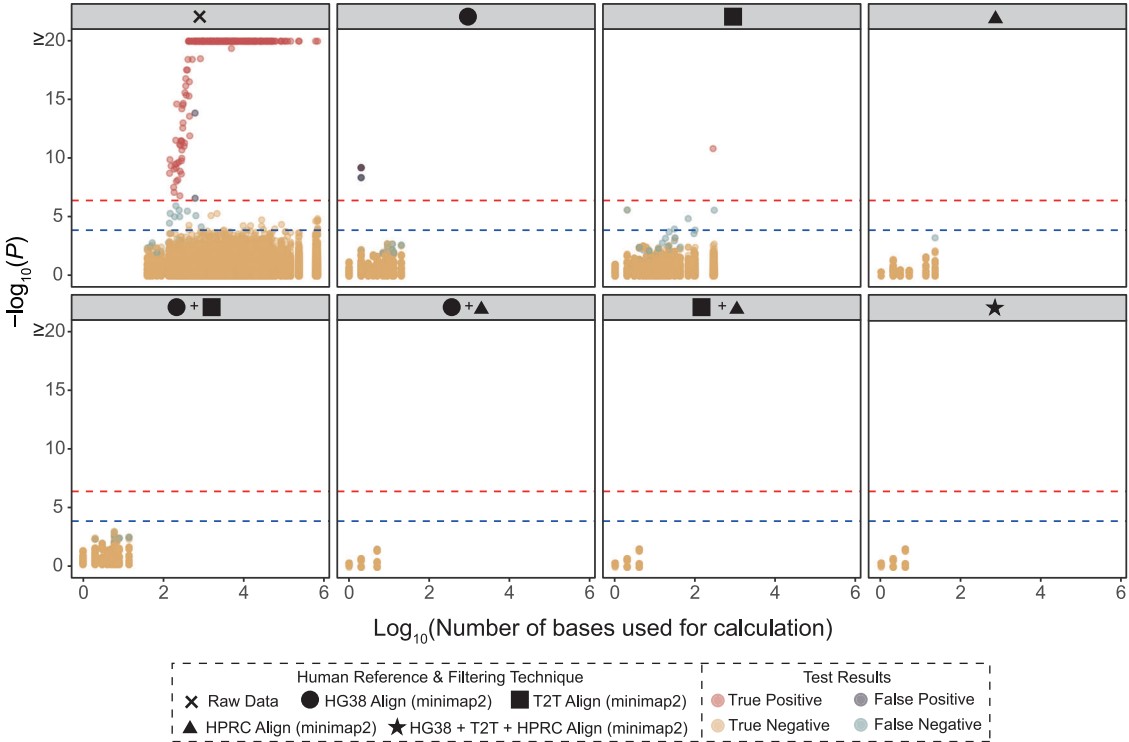

**Fig. 6 | Re-identification from a set of genotype data based on the human reads in fecal samples prevented with improved host filtration.** The 343 fecal samples from Tomofuji et al. Nature Microbiology 2023, with paired genotype data, were re-analyzed with various combinations of updated host filtration methods (GRCh38.p14, T2T-CHM13v2.0, Human Pangenome Reference Consortium 2024 release) resolving host data leakage. The x-axis of the plots indicates the number of bases used for the calculation of the likelihood scores. The y-axis of the plot indicates the two-sided $P$ values calculated using a standard normal distribution based on the standardized likelihood scores. The red and blue dashed lines indicate $p = 4.3 \times 10^{-7}$ (0.05/117,649 tests) and $p = 1.5 \times 10^{-4}$ (0.05/343 tests), respectively. The results of the 117,649 tests (343 genotype data × 343 metagenome data) are indicated as the colors of the points. Some samples could not be used for the re-identification analysis because too few reads remained after filtering, hence the fewer dots shown across host filtration methods. Full description on the calculation of $P$ values can be found in the Methods.

Method 2 having the lowest percentage of reads remaining followed by Method 1, then Method 3.

## Proper host filtration protects against private genomic data leakage

Improper host filtration of metagenomic samples can leak sensitive genomic information. In a recent study, Tomofuji et al.[8] re-identified patients from human reads that leaked through fecal metagenomic data, matching them to blood-derived genotype data from the same individuals (Supplementary Data 1). Their study initially used host filtration steps derived from traditional filtration methods[8]. To test the effectiveness of our approaches to disrupt a host re-identification signal, we applied the above methods to the 343 fecal samples from Tomofuji et al.[8] re-filtering host data with steps outlined in Methods 1 and 2. Using the 343 paired genotype samples to test whether re-identification (from the fecal samples) was still possible (see Methods for details), we found that filtration with any combination of two human references (GRCh38.p14, T2T-CHM13v2.0, HPRC) was sufficient to prevent patient re-identification, haplotype reconstruction, and phenotype prediction (Fig. 6). These data demonstrate the importance of thorough host filtration prior to public upload of mNGS data while providing computationally efficient tools to do so.

## Discussion

While host filtration and host depletion are important steps of careful mNGS analyses, only host filtration has relevant use cases for thousands of already-generated human datasets, especially those initially generated without the original intent of microbial analysis. Through processing whole genome sequencing (WGS) data from metastatic tumor samples, we incidentally identified the impact of insufficient host filtration through artifactual sex biases that human DNA introduced in downstream analyses. These biases had larger impacts on abundance-based metrics and can be mitigated by using qualitative approaches. Nonetheless, such biases introduced by insufficient host filtration likely persist for other metagenomic sequencing datasets, particularly those generated prior to the release of the T2T-CHM13v2.0 reference genome containing the full Y chromosome. Thus, it remains prudent to continue developing and refining techniques to easily scrub human DNA from pre-existing and future mNGS datasets.

Beyond biasing microbial data, an additional consequence of insufficient host filtration in human studies is the retention of personally-identifying human DNA sequences in mNGS datasets. Many mNGS data generation and usage agreements dictate that metagenomic applications of whole genome sequencing will be used to generate and analyze microbial DNA, not human DNA: this is particularly relevant in clinical environments where patients may consent to microbial analysis via mNGS of samples derived from a human host but may not consent to their host genomic content analysis[31]. Moreover, patients are often unaware that untargeted sequencing approaches intended for microbial study will also sequence some amount of host DNA, and failure to comprehensively remove host DNA from metagenomic sequencing data may violate data usage and patient consent agreements by inadvertently enabling deanonymization and reidentification. This is particularly important when depositing microbiome datasets in public repositories, which typically do not require restricted/controlled access[8]. Even when efforts are devoted to preventing re-identification of individuals (for example, by only sharing summary-level data), others have shown that re-identification of participants

from specific GWAS is still possible, or even in DNA mixtures where an individual contributes less than 0.1% of the total genomic material[32]. Therefore, as sequencing methods continue to provide higher mNGS throughput, it is imperative that computationally efficient techniques countering their unintended privacy consequences are made available.

To address this need, we have proposed an efficient, customizable, and effective host filtering pipeline that accurately separates human and microbial reads from mNGS datasets, with demonstrated applications across real samples of varying biomass. We recommend using Method 1, which is time efficient (Fig. 2b) while leaving a majority of the microbial reads (Fig. 3b), but removes enough human reads to disrupt subject reidentification. In cases where the maximum amount of human reads must be removed, even at the cost of losing microbial reads, then Method 2 would be more appropriate. Finally, if a user wants to maximize the remaining microbial reads, then Method 3 would be best. Given the scalability of Movi[27], we anticipate that additional human references can be easily incorporated as they are released without major impacts on runtime, making the approach future proof.

Tomofuji et al. describe the process of performing patient reidentification using relatively few human DNA sequences retained in fecal sequencing data by combining paired genotype data[8]. Using their data, we demonstrated how our methods can prevent patient reidentification, thereby protecting patient privacy. As interest in metagenomic studies of human biological processes grows, increased emphasis should be placed on applying end-to-end privacy-protecting methodologies, inclusive of computational workflows. Sequencing human DNA is a byproduct of mNGS, even in fecal samples, and may persist despite molecular host depletion protocols. Thus, applying computational host filtration techniques remains imperative when performing (or uploading) human-associated microbiome studies.

All host filtration methods remain imperfect due to the underexplored genomic diversity of the human population and the concomitant lack of complete and individualized human reference genomes. Our proposed workflows incorporate more genetic diversity than any computational host filtration approach to date while remaining computationally efficient. Since existing reference databases do not account for the complete set of human genetic variation or non-germline sequence variants frequently found in cancer and other diseases, read-based host filtration approaches may leave a small number of human reads in the data while performing a negative selection for human DNA.

Although removing human reads is an important part of downstream microbial analysis data, this is just one part of the puzzle in properly detecting the true microbial profiles of low biomass human samples. Many other factors, including various ways the sample may become contaminated from the collection process through sequencing, require other tools and strategies beyond our host filtration pipeline to be accounted for.

If researchers desire extra protections to ensure no human reads are inadvertently mapped, a positive selection for microbial reads can be performed using reference databases confirmed to be fully microbial. We conducted this type of analysis using a microbial database scrubbed of human reads derived from Sepich-Poore et al.[7] and similarly noted a resolution of the artifactual sex-difference effect with the cleaned database alone. Additionally, researchers may choose to use a broader range of microbial reference databases beyond RefSeq, which may have more low-complexity regions masked, potentially eliminating some of the mismapping issues leading to sex differences. Additionally, we provide a list to the community of 'false positive taxa' we identified with the addition of T2T-CHM-13v2.0 filtering across common microbial databases (RefSeq release 200 and 210[33], Web of Life version 2[23], Genome Taxonomy Database release 220[34]) as a resource to the community (Supplementary Data 7). However, we caution that cleaning microbial reference databases or using alternative microbial databases in principle cannot address the retention of human reads in

metagenomics datasets due to bias from incomplete representation of variation in the human genome.

Nevertheless, this work highlights the importance of and provides appropriate tools for thorough host filtration to mitigate false alignments and erroneous conclusions. The methods here provide an important starting point for conducting host filtration using state-of-the-art methods while being readily expandable for future improvements and reference databases.

## Methods

All participants from which fecal and skin samples were derived provided informed consent, and protocols were approved by the Institutional Review Boards at the University of Wisconsin-Madison (#2015-0030) and the University of California San Diego (#200844) respectively.

### Human references
Human reference genomes GRCh38.p7, GRCh38.p14, and T2T-CHM-13v2.0 were retrieved from NCBI (Supplementary Data 1). All 94 currently published reference assemblies from the Human Pangenome Reference Consortium website. More information on downloading each reference can be found in Supplementary Data 1.

### HMF data processing
The Hartwig Medical Foundation[15] (HMF) performed DNA sequencing of tumor tissue ($n = 9973$ samples) and mapped reads to reference genome GRCh37.p13 using BWA-MEM[35] (v. 0.7.x) to create BAM files. Pre-aligned BAM files were downloaded from HMF in October 2021, and unmapped reads were extracted from the BAM files. The files were then filtered through fastp[36] (v. 0.20.1) with a length cutoff of 45 bp minimum and default adapter removal, then using minimap2 (v. 2.17) mapped to either GRCh38.p7[37] or GRCh38.p7 + T2T-CHM13v2.0[16] human databases. Finally, samtools[38] (v. 1.11) was used to extract reads which did not align to the human reference. The full command used is: *fastp -l 45 -i $R1 -I $R2 -w 16 --stdout | minimap2 -ax sr -t 16 $human_database - -a | samtools fastq -@ 16 -f 12 -F 256 -1 $R1_out -2 $R2_out*. Following host filtration, reads were aligned to the RefSeq release 200 database using the SHOGUN[39] protocol with Woltka[23] in the Qiita[40] platform. RefSeq[33] release 200 includes the NCBI representative and reference microbial genomes corresponding to release date 2020-05-14. Dimensionality reduction of the corresponding BIOM table[41] was then performed through Gemelli's RPCA function (v. 0.0.6)[18] to create a distance matrix on which PERMANOVA[42] differences across sex and a Robust Aitchison PCA plot were created, both using QIIME2[43] (v. 2022.2.0).

### Parameters used in alignment and indexing host filtration
For all three methods, raw FASTQ files are quality filtered using fastp[36] (v. 0.23.4) with a length cutoff of 45 bp minimum and subject to adapter removal using the full list of adapters hardcoded in fastp (as opposed to relying on fastp's automated detection which is limited to a subset of sequences).

For host filtration methods that utilize sequence alignment, we generated individual minimap2[25] (v. 2.26) indexes using default parameters for each individual human reference genomes. Sequence alignment was then performed with parameters: *-2 -ax sr*. To host filter, sequences were aligned sequentially to each genome discarding reads which mapped. We used samtools[38] (v. 1.19) to reverse unmapped sequences from SAM back into FASTQ format (with arguments *-f 12 -F 256 -N* for paired-end data and *-f 4 -F 256* for single-end data) after each consecutive alignment. Finally, for paired-end data we used fastq-pair[44] (v. 0.4) to sort and match filtered read pairs into individual files.

For host filtration methods that utilize indexing, we generated a full index over all 94 published pangenome references from HPRC release 2023, along with GRCh38.p14 and T2T-CHM-13v2.0 using Movi[27] (unversioned; git commit hash 76d5a6da1ec0aeb0121b5ac7c59b2959

36e23cc1). Movi generates pseudo-matching lengths (PMLs) which are approximations of sequence similarity between the query and the index. We used movi-default to generate PML distributions for each queried read and explored several different mathematical transformations of the resulting PML distributions into singular per-read scores. The PML distributions produced by Movi roughly approximate matching statistics previously validated for sequence classification tasks[27]. Thus, we reasoned that reads with larger PML distribution values had higher similarity to human genomic regions within the index than those with lower values. We tested several transformations of PML distributions into summative scores and devised approach-specific threshold values based on a theoretical human read of length 150 with a singular contiguous matching run of length 31. First, we utilized the maximum PML score within the distribution as the test value and thus computed the threshold for the maximum approach as 31. Next, we calculated the average PML score within the distribution as the test value and thus computed the threshold for the average approach as 3.306. To maximally distinguish PML distributions that feature discontinuous runs of matching nucleotides, we devised a custom metric that magnifies the summative score for read distributions with long stretches of matches above a minimum run-length threshold (denoted $w$). In concordance with the previous maximum and average metrics, we computed the threshold for the custom approach as 0.175 with a minimum run-length value of 5. Through experimental validation on mixed human/microbe datasets, we found that our custom metric had the best discriminative performance. To verify this approach, we ran a grid search over various thresholds for the custom metric (thresholds: 0.145 to 0.200 by increments of 0.005; minimum run-length: 2 to 12 by increments of 1) on our simulated data for which we had labeled ground truth (see "Simulated data"). The results showed strong recall for our use cases at the default threshold, but we acknowledge that some users may prefer higher stringency on human DNA removal even at the cost of inadvertent microbial DNA removal. Thus, we implemented both the metric ("maximum", "average", or "custom") and the numerical threshold as configurable options in our host filtration pipeline to meet the needs of all users. The equation for the custom metric is denoted below:

$$\frac{1}{2L}(\max(PML\ distribution) + \left(\left(\sum_{r \in R} r\right) \cdot \log(|R| + 1)\right)) where\ r \in R\ if\ len(r) > w \quad (1)$$

### Host filtration benchmarking data processing

We benchmarked three methods to compare different combinations of the aforementioned alignment-based and indexing-based host filtration approaches. We use minimap2 for alignment based on its support for this application in prior work[11].

**Method 1**. Step i (filled circle): aligned reads to human reference GRCh38.p14 with minimap2 (v. 2.26), then used samtools (v. 1.19) to extract reads that did not align to the human reference. Step ii (filled square): aligned remaining reads from step i to human reference T2T-CHM13v2.0 with minimap2 (v. 2.26), then used samtools (v. 1.19) to extract reads that did not align to the human reference. Step iii (star): matched remaining reads from step ii to an aggregated human reference set consisting of GRCh38.p14, T2T-CHM13v2.0, and the 94 HPRC pangenomes using indexing-based filtration with Movi, as described above.

**Method 2**. Step i (filled star): aligned reads sequentially to GRCh38.p14, T2T-CHM13v2.0, and the 94 HPRC pangenomes with minimap2 (v. 2.26), then used samtools (v. 1.19) to extract reads that did not align with each iteration. Step ii (star): matched remaining reads from step i to an aggregated human reference set consisting of

GRCh38.p14, T2T-CHM13v2.0, and the 94 HPRC pangenomes using indexing-based filtration with Movi, as described above.

**Method 3**. Step i (star): matched reads to an aggregated human reference set consisting of GRCh38.p14, T2T-CHM13v2.0, and the 94 HPRC pangenomes using indexing-based filtration with Movi, as described above.

Although all Methods use GRCh38.p14, T2T-CHM13v2.0, and HPRC as part of the Movi indexing approach, we want to emphasize that only Method 2 also uses them for minimap2-based alignment. Although minimap2 and Movi are both useful tools for linking short reads to reference genomes, their underlying algorithms are distinct and thus the two tools produce differing results (Fig. 3) and runtimes (Fig. 2b). Minimap2 uses a traditional seed-chain-align approach to identify exact matches to a reference, while Movi uses the Move structure introduced by Nishimoto and Tabei in 2021 to compute pseudo-matching lengths to a reference[45]. Combining minimap2 and Movi utilizes the respective strengths of each of their implementations, and appears to result in the highest number of reads removed.

We calculated the number of microbial, bacterial, eukaryota, viral, and archaea genome frequencies listed in Supplementary Data 6 using the SHOGUN[39] protocol via the Qiita[40] platform with the RefSeq release 210 (2022-01-01) database. We listed forward and reverse read counts separately in Supplementary Data 6 to directly report the number of reads mapped to microbial taxa. Read counts listed throughout all other portions of the manuscript count forward and reverse reads as a single count. Additionally, in the case of the Tissue Samples from Various Metastatic Cancer, the 'Before host filtration number' is following GRCh38.p7 host filtration and not the purely raw reads as in the other cases.

As a resource to the community, we calculate microbial alignments from reads retained following GRCh38.p14 filtration, but removed with the addition of T2T-CHM-13v2.0 filtration. We do this by tabulating the additional reads removed from the aforementioned 100 colorectal tissue tumor samples from HMF when incorporating T2T-CHM-13v2.0 for host filtration (Method 1 step ii) referred to as "T2T-filtered" reads above. We aligned these removed reads to a diverse set of microbial reference databases – RefSeq release 200[33], RefSeq release 210[33], Web of Life release 2[23], and GTDB release 220[22] – and reported the resulting spurious microbial alignments (Supplementary Data 7).

### Statistics and reproducibility

We used two-sided Wilcoxon signed-rank tests using SciPy[46] (v. 1.8.0) to assess differences in medians between reads retained or removed across differing methods. Because this test involves ranking the absolute differences between pairs, and since reads retained or reads removed tend to either decrease or increase respectively across pairs, the resulting test statistic reported is 0 for all comparisons. For all boxplots, throughout the figures, the box represents the interquartile range (IQR), with the centerline being the median and the top and bottom of the box representing Q1 and Q3. Boxplots were generated using matplotlib[47] (v. 3.8.0), and the whiskers of the plot were left at matplotlib defaults, making them +/−1.5 from the IQR. Outliers were removed from the boxplots since a scatter plot with all dots was overlaid. To facilitate computationally efficient benchmarking of the host filtration methods, we analyzed a subset of $n = 100$ samples from the HMF dataset, and $n = 50$ samples from the Alzheimer's disease fecal dataset while using complete datasets for all other sample types. All $p$ values are rounded two decimal places, and all test statistics are reported with three significant figures unless otherwise specified.

### Simulated data

Data was simulated using ART Illumina[46,48] (v. 2.5.8). Human reads were simulated using HPRC genomes, and microbial reads were simulated

from FDA-ARGOS[28]. Supplementary Data 1 further describes simulated dataset accession.

**Re-identification analysis with updated host filtration methods**

For the re-identification analysis, we utilized human reads extracted from mNGS data and imputed SNP array data in the previous study[8]. The main steps in the human read extraction were as follows: (i) trimming of low-quality bases, (ii) identification of candidate human reads, (iii) removal of duplicated reads, and (iv) removal of the potential bacterial reads. We trimmed the raw reads to clip Illumina adapters and cut off low-quality bases using the Trimmomatic[49] (v. 0.39; parameters: ILLUMINACLIP:TruSeq3-PE-2.fa:2:30:10:8:true TRAILING:20 MINLEN:60). We discarded reads less than 60 bp in length after trimming. Then, we mapped the trimmed reads to the human reference genome (GRCh37, human_g1k_v37_decoy) using bowtie2[50] (v. 2.3.5.1) with the '−no-discordant' option and retained only the properly mapped reads. Next, we performed duplicate removal by Picard MarkDuplicates (v. 2.22.8) with 'VALIDATION_STRINGENCY = LENIENT' option. Finally, we mapped the duplicate removed reads to the bacterial reference genome set constructed in Kishikawa et al.[51]. This reference was composed of the 7881 genomes including those derived from Nishijima et al.[52] and those identified in the cultivated human gut bacteria projects[53–55]. We kept only reads of which both paired ends failed to align. The resulting reads were defined as human reads and used in the subsequent analyses. Then, extracted human reads were subjected to the host filtration methods, namely the first steps two steps of Method 1: GRCh38.p14 alignment, and GRCh38.p14/T2T-CHM13v2.0 alignment, along with the first step of Method 2: GRCh38.p14/T2T-CHM13v2.0/HPRC alignment.

For the re-identification analysis, we utilized likelihood score-based method introduced in the previous study[8]. We calculated the likelihood that each sample in the genotype dataset produced the observed human reads in the fecal samples from two input data; (i) human reads in the gut-derived mNGS data which were mapped to the human reference genome and (ii) genotype dataset of the multiple samples. We extracted the SNP sites which were covered by at least a read and included in the reference panel by 'bcftools mpileup'[56] with the '-T' option. To get independent SNP sites, we applied clumping to the list of the SNPs which were covered by at least a read. We used '−indep-pairwise 100 30 0.1' option in PLINK for clumping at Rsq = 0.1. Then, we calculated the likelihood according to the model proposed in Li et al.[57]. Suppose an SNP site $i$ was covered by $n_i$ reads in the gut-derived mNGS data, $k_i$ reads were from the reference allele, and $n_i − k_i$ reads were from the alternative allele. bcftools[56] (v. 1.10.2) was used to calculate the read coverage with '-q 40 -Q 20' options. The error probability of the read bases was $\varepsilon$ and error independency was assumed. In this study, $\varepsilon$ was set at $1 \times 10^{-6}$ following the assumption in Li et al.[57]. At the SNP site $i$, the number of the alternative allele of an individual $j$ ($g_{i,j}$) could be 0 (Ref / Ref), 1 (Ref / Alt), or 2 (Alt / Alt). Then, the likelihood that the sample with a $g_i$ alternative alleles at SNP site $i$ produced the observed human reads in the gut-derived mNGS data was expressed as

$$L_{i,j}(g_{i,j}, n_i, k_i) = \frac{1}{2^{n_i}} [(2 - g_{i,j})\varepsilon + g_{i,j}(1 - \varepsilon)]^{n_i - k_i} [(g_{i,j}\varepsilon + (2 - g_{i,j})(1 - \varepsilon)]^{k_i}$$

(2)

When the clumping procedure retained $N$ independent SNP sites, a log-transformed likelihood (likelihood score; *LS*) that a genotype data produced the observed human reads in the gut-derived mNGS data was expressed as

$$LS_j = \sum_{i=1}^{N} \log\left(L_{i,j}\left(g_{i,j}, n_i, k_i\right)\right)$$

(3)

Next, we drew the background distribution of the likelihood score from (i) human reads in the gut-derived mNGS data which were mapped to the human reference genome, and (ii) allele frequency data for the SNP sites used for calculating the likelihood score. In this study, Japanese subjects in the combined reference panel of 1KG Project Phase 3[58] version 5 genotype ($n = 104$) and Japanese WGS data ($n = 1037$) were used to calculate the allele frequency[59]. When an alternative allele frequency at SNP site $i$ was $p_i$ and the number of the alternative allele was $g_{i,pop}$ (=0, 1, or 2), theoretical genotype frequencies at SNP site $i$ were expressed as

$$P\left(g_{i,pop}, p_i\right) = \begin{cases} (1 - p_i)^2, & (g_{i,pop} = 0) \\ 2p_i(1 - p_i), & (g_{i,pop} = 1) \\ p_i^2, & (g_{i,pop} = 2) \end{cases}$$

(4)

Then, the expected log transformed likelihood that a genotype data randomly drawn from the specified population produced the observed human reads in the mNGS data was expressed as

$$E\left(LS_{pop}\right) = \sum_{i=1}^{N} E\left(LS_{i,pop}\right) = \sum_{i=1}^{N} \sum_{g_{i,pop}=0}^{2} P\left(g_{i,pop}, p_i\right) \log(L_i(g_{i,pop}, n_i, k_i))$$

(5)

Given that SNP sites were independent, the variance of the likelihood score in a specific population was expressed as

$$V\left(LS_{pop}\right) = \sum_{i=1}^{N} V\left(LS_{i,pop}\right)$$
$$= \sum_{i=1}^{N} \sum_{g_{i,pop}=0}^{2} P\left(g_{i,pop}, p_i\right) \left[\log(L_i\left(g_{i,pop}, n_i, k_i\right)) - E\left(LS_{i,pop}\right)\right]^2$$

(6)

Using $E\left(LS_{pop}\right)$ and $V\left(LS_{pop}\right)$, we calculated the standardized likelihood score of the individual $j$ as $\frac{(LS_j - E(LS_{pop}))}{\sqrt{V(LS_{pop})}}$. We transformed standardized likelihood scores to $P$ values based on the normal distribution. We identified the pair of the gut-derived mNGS and genotype data (imputed SNP array data was used in this study) derived from the same individuals based on the $P$ values.

**Reporting summary**

Further information on research design is available in the Nature Portfolio Reporting Summary linked to this article.

# Data availability

The raw HMF data used in this study is under the purview of the Hartwig Medical Foundation and contains patient-protected information that cannot be shared publicly (see https://hartwigmedical.github.io/documentation/data-access-request-methods.html for data access guidelines for access request details). The FDA-ARGOS database used in this study is publicly available from the official website (https://argos.igs.umaryland.edu) as well as NCBI BioProject PRJNA231221 (https://www.ncbi.nlm.nih.gov/bioproject/231221). The human exome data used in this study is derived from the IGSR phase 3 data, which is available via the official EMBL-EBI portal (https://www.internationalgenome.org/data). The atopic dermatitis skin sample data and the Alzheimer's disease fecal sample data used in this study are available from ENA under accession PRJEB83637. The fecal sample data from Tomofuji et al. are publicly available from JGA under accessions JGAS000260, JGAS000316, and JGAS000531 (https://www.ddbj.nig.ac.jp/jga/index-e.html). The blood sample data from Tomofuji et al. are publicly available from EGA under accession EGAS00001007027.

## Code availability

Code and instructions to implement the methods of host filtration can be found on GitHub (https://github.com/cguccione/human_host_filtration). Code used to create simulated data, run host filtration metrics and create figures can be found on GitHub (https://github.com/cguccione/host-filtration-notebooks).

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

## Acknowledgements

This work was supported by AGA Research Foundation (AGA Research Scholar Award AGA2022-13-05) and NIH grant R01 CA270235 to K.C. The study was supported in part by the NIDDK-funded San Diego Digestive Diseases Research Center (P30 DK120515) to K.C. Additionally this work was supported by NIH grants (R01 CA241728, P30 CA023100, NIH/NIGMS T32GM007198, NIH Pioneer DP1AT010885), the National Cancer Institute (NCI U24CA248454), and CDC award 75D301-22-C-14717 to R.K. The study was supported in part by R21HG013433 to B.L. This study was supported in part by JSPS KAKENHI (22H00476), and AMED (JP24km0405217, JP24ek0109594, JP24ek0410113, JP24kk0305022, JP243fa627002, JP243fa627010, JP243fa627011, JP24zf0127008, JP24tm0524002, JP24wm0625504, JP24gm1810011), JST Moonshot R&D (JPMJMS2021, JPMJMS2024), to Y.O., with additional support from Takeda Science Foundation, Ono Pharmaceutical Foundation for Oncology, Immunology, and Neurology, Bioinformatics Initiative of Osaka University Graduate School of Medicine, Institute for Open and Transdisciplinary Research Initiatives, Center for Infectious Disease Education and Research (CiDER), and Center for Advanced Modality and DDS (CAMaD), Osaka University. This project was enabled in part by the Alzheimer's Gut Microbiome Project (AGMP), supported by the National Institute on Aging grants: 1U19AG063744 and 3U19AG063744-04S1, awarded to Dr. Kaddurah-Daouk at Duke University in partnership with multiple academic institutions. As such, the investigators within the AGMP not listed in this publication's authors' list, provided analysis-ready data, but did not participate in designing the study, conducting the analyses or writing of this manuscript. A listing of AGMP Investigators can be found at https://alzheimergut.org/meet-the-team/. A complete listing of the AD Metabolomics Consortium (ADMC) investigators can be found at: https://sites.duke.edu/adnimetab/team/. We thank Cameron Martino for his support and advice throughout this project.

## Author contributions

C.G. and L.P. conceived and designed the study. C.G., L.P., A.G., Y.C., A.H.D., G.H., T.N., and R.L.G. collected and processed the data. C.G., L.P., D.M., and N.D. performed the primary analyses. Y.T., K.S., and Y.O. performed the re-identification analyses. M.Z. and B.L. conceptualized, implemented, and supported application of the pangenome index. C.G. and L.P. performed the statistical analysis. D.M., G.D.S.P., and S.E.B., aided in interpreting the results and drafting the manuscript. C.G. and L.P. wrote the manuscript with input from all authors. K.C. and R.K. supervised the project. All authors reviewed and approved the final manuscript.

## Competing interests

D.M. is a consultant for BiomeSense, Inc., has equity and receives income. The terms of these arrangements have been reviewed and approved by the University of California, San Diego in accordance with its conflict of interest policies. G.H. is the recipient of the Robert A. Winn Diversity in Clinical Trials: Career Development Award, which is partly funded by Bristol-Meyer Squibb Foundation. B.L. is the owner of InOrder Labs LLC. K.C. has research grant support from Phathom Pharmaceuticals. R.K. is a scientific advisory board member, and consultant for BiomeSense, Inc., has equity and receives income. He is a scientific advisory board member and has equity in GenCirq. He is a consultant for DayTwo, and receives income. He has equity in and acts as a consultant for Cybele. He is a co-founder of Biota, Inc., and has equity. He is a cofounder of Micronoma, and has equity and is a scientific advisory board member. The terms of these arrangements have been reviewed and approved by the University of California, San Diego in accordance with its conflict of interest policies. The remaining authors declare no competing interests.

## Additional information

[1]Division of Biomedical Informatics, Department of Medicine, University of California San Diego, La Jolla, CA, USA. [2]Bioinformatics and Systems Biology Program, University of California San Diego, La Jolla, CA, USA. [3]Department of Pediatrics, University of California San Diego, La Jolla, CA, USA. [4]Medical Scientist Training Program, University of California, San Diego, La Jolla, CA, USA. [5]Department of Genome Informatics, Graduate School of Medicine, the University of Tokyo, Tokyo 113-8654, Japan. [6]Department of Statistical Genetics, Osaka University Graduate School of Medicine, Suita 565-0871, Japan. [7]Laboratory for Systems Genetics, RIKEN Center for Integrative Medical Sciences, Yokohama 230-0045, Japan. [8]Shu Chien-Gene Lay Department of Bioengineering, University of California San Diego, La Jolla, CA, USA. [9]Feinberg School of Medicine, Northwestern University, Chicago, IL, USA. [10]Department of Computer Science, Johns Hopkins University, Baltimore, MD, USA. [11]Biomedical Sciences Graduate Program, University of California San Diego, La Jolla, CA, USA. [12]Halıcıoğlu Data Science Institute, University of California San Diego, La Jolla, CA, USA. [13]Department of Cognitive Science, University of California San Diego, La Jolla, CA, USA. [14]Weill Institute for Neurosciences. Department of Neurology. University of California, San Francisco (UCSF), San Francisco, CA, USA. [15]Department of Dermatology, University of California San Diego, La Jolla, CA, USA. [16]Rady Children's Hospital, San Diego, CA, USA. [17]Center for Microbiome Innovation, University of California San Diego, La Jolla, CA, USA. [18]Laboratory of Statistical Immunology, Immunology Frontier Research Center (WPI-IFReC), Osaka University, Suita 565-0871, Japan. [19]Premium Research Institute for Human Metaverse Medicine (WPI-PRIMe), Osaka University, Suita 565-0871, Japan. [20]VA San Diego Healthcare System, San Diego, CA, USA. [21]Moores Cancer Center, University of California San Diego, La Jolla, CA, USA. [22]Department of Computer Science and Engineering, University of California San Diego, La Jolla, CA, USA. [23]These authors contributed equally: Caitlin Guccione, Lucas Patel. [24]These authors jointly supervised this work: Kit Curtius, Rob Knight. ✉e-mail: kcurtius@health.ucsd.edu; robknight@ucsd.edu

