## [Transparent Peer Review file · Nature Communications]

Incomplete human reference genomes can drive false sex biases and expose patient-identifying information in metagenomic data

Corresponding Author: Dr Rob Knight

Version 0:

Reviewer comments:

Reviewer #1

(Remarks to the Author)

The authors have done a truly stellar job at revising their manuscript in light of my comments. I really enjoyed reading their extensive new analysis and thoughtful responses, especially relaxing their statements where I found them a bit strong (e.g., re the blood microbiome, various "recommended" workflows, etc). I think the manuscript is improved and a valuable contribution to the field, and I recommend its acceptance.

I really appreciate the analysis of which taxa are soaking up human reads. Their results -- especially re: *T gondii*, are so important for the metagenomic community, and I would almost love to see a paragraph in the discussion on avoiding these common "false positive" taxa. I regularly see papers published where suspect organisms (specifically *T gondii* and other similar bugs) are reported in abundance tables without comment, when they are pretty clearly misalignments. The authors have somewhat unintentionally crafted a resource list of organisms like this, and it would benefit the field if that fact were called out in the text succinctly. But, of course, it might be out of scope and best done in a different paper.

I prefer to sign my reviews when possible -- Braden Tierney

(Remarks on code availability)

Reviewer #2

(Remarks to the Author)

This is a resubmitted and improved version of a manuscript that I had previously reviewed for Nature Medicine.

I feel the authors have improved the manuscript in this new version and answered positively to my previous comments.

(Remarks on code availability)

Response letter to Reviewer comments - December 2024

Note: Below, the reviewer comments are in **black**, our responses to reviewers are interleaved in **blue** and whenever a direct quote from the paper is made, colored in **red**, and referenced by line number to the text.

Reviewer #1

The authors have done a truly stellar job at revising their manuscript in light of my comments. I really enjoyed reading their extensive new analysis and thoughtful responses, especially relaxing their statements where I found them a bit strong (e.g., re the blood microbiome, various "recommended" workflows, etc). I think the manuscript is improved and a valuable contribution to the field, and I recommend its acceptance.

I really appreciate the analysis of which taxa are soaking up human reads. Their results -- especially re: *T gondii*, are so, so important for the metagenomic community, and I would almost love to see a paragraph in the discussion on avoiding these common "false positive" taxa. I regularly see papers published where suspect organisms (specifically *T gondii* and other similar bugs) are reported in abundance tables without comment, when they are pretty clearly misalignments. The authors have somewhat unintentionally crafted a resource list of organisms like this, and it would benefit the field if that fact were called out in the text succinctly. But, of course, it might be out of scope and best done in a different paper.

I prefer to sign my reviews when possible -- Braden Tierney

Author response:

We thank Dr. Tierney very much for the positive feedback on our manuscript. We agree that avoiding common "false positive taxa" is essential for proper microbial analysis and have incorporated these results further into the manuscript. We do agree that a further manuscript that highlights this specific issue citing this paper in a paper with a title/abstract that is mostly about that topic and will be easily findable by researchers is a great idea, and will look into doing that as a subsequent piece of work. In the context of the present manuscript, we created a table (Supplemental Table 7) that contains microbial taxa found by mapping reads filtered following inclusion of T2T-CHM13v2.0, but not with GRCh38.p14 onto RefSeq release 200, RefSeq release 210, Web of Life release 2 and the Genome Taxonomy Database release 220. This complements and extends Supplementary Table 3 by exploring several additional microbial reference databases, but reports the total alignments following bowtie2 (rather than the total microbial hits following Woltka) to more generally reflect the expected conditions for researchers relying on alignment-based microbial quantification. We refer to this table in both the discussion and the methods of the manuscript with the following additional text:

L 412-415: Additionally, we provide a list to the community of 'false positive taxa' we identified with the addition of T2T-CHM-13v2.0 filtering across common microbial databases (RefSeq release 200 and 210, Web of Life version 2, Genome Taxonomy Database release 220) as a resource to the community (Supplementary Table 7).

L 543-550: As a resource to the community, we calculate microbial alignments from reads retained following GRCh38.p14 filtration, but removed with the addition of T2T-CHM-13v2.0 filtration. We do this by tabulating the additional reads removed from the aforementioned 100 colorectal tissue tumor samples from HMF when incorporating T2T-CHM-13v2.0 for host filtration (Method 1 step ii) referred to as "T2T-filtered" reads above. We aligned these removed reads to a diverse set of microbial reference databases – RefSeq release 200, RefSeq release 210, Web of Life release 2, and GTDB release 220 – and reported the resulting spurious microbial alignments (Supplementary Table 7).

Reviewer #2

This is a resubmitted and improved version of a manuscript that I had previously reviewed for Nature Medicine.

I feel the authors have improved the manuscript in this new version and answered positively to my previous comments.

Author response:

We thank the Reviewer for their positive feedback.

Editors

In line with Reviewer #1, we -the editors- feel that it will be great if you could add a brief discussion and a list of the identified false positive taxa as a resource for the community (perhaps in a Supplementary Table or via a Github or Zenodo link?)

Author response:

As described in the response to Reviewer #1, we have added text to both the discussion and methods sections describing the identified false positive taxa, and added a new supplemental table 7 which lists the taxa.